# Fabrication of Nanoformulation Containing Carvedilol and Silk Protein Sericin against Doxorubicin Induced Cardiac Damage in Rats

**DOI:** 10.3390/ph16040561

**Published:** 2023-04-07

**Authors:** Mohammad Shariq, Tarique Mahmood, Poonam Kushwaha, Saba Parveen, Arshiya Shamim, Farogh Ahsan, Tanveer A. Wani, Seema Zargar, Rufaida Wasim, Muhammad Wahajuddin

**Affiliations:** 1Department of Pharmacy, Integral University, Lucknow 226026, Uttar Pradesh, India; mhmshariq@iul.ac.in (M.S.); poonam@iul.ac.in (P.K.); sabapar@iul.ac.in (S.P.); arshiyas@iul.ac.in (A.S.); farogh@iul.ac.in (F.A.); rufaidaw@student.iul.ac.in (R.W.); 2Department of Pharmaceutical Chemistry, College of Pharmacy, King Saud University, P.O. Box 2457, Riyadh 11451, Saudi Arabia; twani@ksu.edu.sa; 3Department of Biochemistry, College of Science, King Saud University, P.O. Box 2452, Riyadh 11451, Saudi Arabia; szargar@ksu.edu.sa; 4Institute of Cancer Therapeutics, School of Pharmacy and Medical Sciences, Faculty of Life Sciences, University of Bradford, Bradford BD7 IDP, UK; m.wahajuddin@bradford.ac.uk

**Keywords:** carvedilol, sericin, doxorubicin, cardioprotective

## Abstract

Nanotechnology has emerged as an inspiring tool for the effective delivery of drugs to help treat Coronary heart disease (CHD) which represents the most prevalent reason for mortality and morbidity globally. The current study focuses on the assessment of the cardioprotective prospective ofanovel combination nanoformulation of sericin and carvedilol. Sericin is a silk protein obtained from *Bombyx mori* cocoon and carvedilol is a synthetic nonselective β-blocker. In this present study, preparation of chitosan nanoparticles was performed via ionic gelation method and were evaluated for cardioprotective activity in doxorubicin (Dox)-induced cardiotoxicity. Serum biochemical markers of myocardial damage play a substantial role in the analysis of cardiovascular ailments and their increased levels have been observed to be significantly decreased in treatment groups. Treatment groups showed a decline in the positivity frequency of the Troponin T test as well. The NTG (Nanoparticle Treated Group), CSG (Carvedilol Standard Group), and SSG (Sericin Standard Group) were revealed to have reduced lipid peroxide levels (Plasma and heart tissue) highly significantly at a level of *p* < 0.01 in comparison with the TCG (Toxic Control Group). Levels of antioxidants in the plasma and the cardiac tissue were also established to be within range of the treated groups in comparison to TCG. Mitochondrial enzymes in cardiac tissue were found to be elevated in treated groups. Lysosomal hydrolases accomplish a significant role in counteracting the inflammatory pathogenesis followed by disease infliction, as perceived in the TCG group. These enzyme levels in the cardiac tissue were significantly improved after treatment with the nanoformulation. Total collagen content in the cardiac tissue of the NTG, SSG, and CSG groups was established to be highly statistically significant at *p* < 0.001 as well as statistically significant at *p* < 0.01, respectively. Hence, the outcomes of this study suggest that the developed nanoparticle formulation is effective against doxorubicin-induced cardiotoxicity.

## 1. Introduction

In the current scenario of therapeutics, nanotechnology has emerged as a promising technology for effective and targeted delivery along with many other beneficial features such as dose reduction, controlled release, minimal side effects, low dose frequency, and enhanced patient compliance [1,2]. There are numerous synthesis processes for a variety of nanoparticles after several decades of rigorous research. Although the complexity of nanoparticle synthesis has increased significantly over the years, our knowledge of how nanoparticles develop still lags far behind [3].

Cardiovascular ailments cause around three million fatalities in India each year, accounting for 25% of all deaths. Furthermore, research on Asian Indians residing abroad reveals a 40% higher incidence of ischemic heart disease (IHD) than the European population [4].

Cardiotoxicity is defined by cardiac dysfunctions, arrhythmia, hypotension, achypnoea, oedema, heart muscle injury, alterations in transmission pathways, and toxic effects on the heart [5]. Cardiomyopathy is characterized by muscle damage to heart tissues rendering the cardiac muscleincapable of efficient pumping leading to heart failure [6,7]. Cardiac hypertrophy is a compensatory response to volume or pressure stress, transmutations of sarcomeric and other proteins, or a decrease in contractile mass right before myocardial infarction. Ventricular hypertrophy is reported as a potentially fatal progression of hemodynamically stressful disorders such as valve dysfunction as well as hypertension. The pathogenesis of hypertrophy is complicated, involving several molecular and cellular systems. The exploration of the molecular mechanisms of myocardial hypertrophy presents a significant undertakingto protect cardiac tissue from pathological remodelling or to place a stop to the anticipated trend toward heart failure [8].

Doxorubicin is an anthracycline class antitumor drug which has the potential for producing cardiotoxicity [9,10]. Acute and persistent cardiomyopathy may result from doxorubicin-induced cardiotoxicity. Chronic DOX-induced cardiac toxicity is dose-dependent, whereas acute cardiotoxicity develops after receiving a large dose and may cause immediate tachyarrhythmia as well as acute heart failure. In such circumstances, the patient may experience dilated cardiomyopathy even years after finishing all DOX treatments. Acute and long-term DOX exposure can cause cardiac toxicity that can result in cardiomyopathy, cardiac dysfunction, severe heart failure, and even death [11]. The rationale for the selection of the two active ingredients was to achieveimproved cardioprotective efficacy, as sericin is a natural silk protein with proven cardioprotective effects and has long been used in traditional systems of medicine. In addition, carvedilol is a non-selective β-blocker and is helpful in cardiac diseases and hypertension. To cope with the low aqueous solubility and oral bioavailability issues of carvedilol, a nanoparticle formulation was developed [12,13]. The current study focuses on the evaluation of the cardioprotective potential of a novel combination nanoformulation of sericin and carvedilol.

## 2. Results

### 2.1. Heart Weight Body Weight Ratio

The heart:body weight ratio is a numerical value that can be obtained by dividing the total weight of the heart by the total weight of the body; the results are illustrated in Figure 1.

### 2.2. Biochemical Estimations (Serum)

The analysis of cardiovascular disorders heavily relies on biochemical indicators of myocardial injury. In the blood serum, many enzymatic parameters were estimated. Biochemical estimates were made from the collected serum, and the results indicate that the toxic control group had an increase in AST, ALP, ALT, CK, CK-MB, and LDH that was statistically highly significant (^a^
*p*< 0.001). In the treatment groups, it was noted that the elevated levels of AST, ALP, ALT, CK, CK-MB, and LDH were dramatically lowered at a significance level of ^b^
*p* < 0.01, and ^c^
*p* < 0.05. Comparing the per se group to the NCG, no discernible difference was seen. In blood serum, many enzymatic parameters were estimated. Figure 2 displays the outcome.

### 2.3. Troponin T

The various groups were investigated for the occurrence of troponin T and found that no animals in the NCG and the per se group revealed a positive consequence for troponin T and all the animals in the toxic control groups showed the presence of troponin T. One animal each in CSG and SSG showed the presence of troponin T while no animal in NTG showed the existence of troponin T. The outcomes for positive and negative results are indicated in Figure 3.

### 2.4. Lipid Peroxide Marker (Plasma)

Significant information on lipid peroxidation after myocardial damage is provided by the plasma’s lipid peroxidative parameters. When oxygen is present, a process known as lipid peroxidation causes lipids to break down. Cellular damage results from the free radicals stealing electrons from the cell membrane during this process. As a result, the markers that are released during lipid peroxidation were estimated. It was discovered that the levels of TBARS, lipid hydroxy peroxidase, and conjugated dienes were considerably higher in the toxic control group at a significance level of ^a^
*p* < 0.0001. The NTG, CSG, and SSG groups demonstrated dramatically decreased levels of lipid peroxide at a significance level of ^b^
*p* < 0.001. The Per se group did not vary from the NCG in any way. The outcome is displayed in Figure 4.

All data were reported as mean ± Standard deviation (*n* = 6). Where Treatment groups (NTG, CSG, and SSG) were compared to TCG, while TCG and PNG were compared to Normal Control, and the levels of significance were observed as ^a^
*p* < 0.0001, ^b^
*p* < 0.001 and, ^c^
*p* < 0.01

### 2.5. Lipid Peroxide Marker (Heart Tissue)

The concentrations of conjugated dienes and TBARS Lipid hydroxy peroxidase were examined in heart tissue homogenate. TBARS Lipid hydroxyperoxidase and conjugated dienes levels were found to be highly significantly increased at a level of ^a^
*p* < 0.0001 in the toxic control group. The NTG, CSG, and SSG groups showed significantly reduced lipid peroxide levels at ^b^
*p* < 0.001. The Perse group did not show any difference from the NCG rats. The results are shown in Figure 5.

All data were reported as mean ± Standard deviation (*n* = 6). Treatment groups (NTG, CSG, and SSG) were compared to TCG, while TCG and PNG were compared to Normal Control, and the levels of significance were observed as ^a^
*p* < 0.0001, ^b^
*p* < 0.001, and ^c^
*p* < 0.05. TBARS, lipid hydroperoxides, and conjugated dienes values were expressed in nmol/mg, mmol/100 g tissue, and mmol/100 g tissue, respectively. Log_10_ was utilized for the scale of the *Y*-axis.

### 2.6. Non-Enzymatic Antioxidant Marker (Plasma)

The measurement of non-enzymatic factors found in plasma reveals the severity of myocardial damage. Vitamin C, Vitamin E, and Glutathione levels significantly increased at a level of ^a^
*p* < 0.0001 in the toxic control group, while significantly decreased ^b^
*p* < 0.001 in the NTG, CSG, and SSG groups. When compared to the NCG, the Perse group did not exhibit any discernible difference. Figure 6 displays the outcomes.

### 2.7. Non-Enzymatic Antioxidant Marker (Heart Tissue)

The estimation of non-enzymatic parameters present in the tissue indicates the extent of injury in the myocardium. Figure 7 displays the findings.

### 2.8. Enzymatic Antioxidant Marker (Heart Tissue)

The functions of the following enzymes were measured in the heart: catalase (CAT), superoxide dismutase (SOD), glutathione reductase (GR), glutathione peroxidase (GPx), and glutathione-S-transferase (GST). The toxic control group had significantly lower levels of antioxidant indicators at ^a^
*p* < 0.0001, which were then restored in the treatment groups at levels of significance at ^b^
*p* < 0.001. Figure 8 presents these findings.

### 2.9. Mitochondrial Enzymes (Heart Tissue)

Isocitrate Dehydrogenase (IDH), -Ketoglutarate Dehydrogenase (KDH), Malate Dehydrogenase (MDH), and Succinate Dehydrogenase (SDH) were only a few of the metabolic enzymes that were assessed in the various treated heart groups. The decline in levels of mitochondrial enzymes was highly significantat a level of ^a^
*p* < 0.0001 and ^b^
*p* < 0.001 in the toxic group, whereas the nanoparticle treatment group showed significant elevation at a level of ^b^
*p* < 0.001. The outcomes are displayed in Figure 9.

### 2.10. Lysosomal Hydrolases (Heart Tissue)

Enzymes called lysosomal hydrolases play a crucial role in the inflammatory process. Lysosomal hydrolase activity is enhanced during doxorubicin-induced MI, which may be the cause of the injured heart and tissue. Myocardial cell or tissue membrane stabilization, especially of the lysosomal membranes, is hypothesized to prolong the life of ischemic heart muscle and aid in the prevention of MI. The levels of lysosomal hydrolases were highly significantly declined in the toxic group at a level of ^a^
*p* < 0.0001, which upon nanoparticle treatment were highly significantly elevated at a level of ^a^
*p* < 0.0001. The outcomes are displayed in Figure 10.

### 2.11. Total Collagen Content

The total collagen content was analyzed in the heart tissue samples by analyzing the hydroxyl proline content and then converting it to collagen content by multiplying it with the conversion factor. Collagen content levels in the toxic control group showed an increase at a significance level of ^a^
*p* < 0.0001. The NTG, CSG, and SSG groups showed a remarkable reduction in collagen content in the tissue at significance levels of ^b^
*p* < 0.001 and ^c^
*p* < 0.01 as compared to the toxic group. When compared with the NCG, the Perse group showed no variation in collagen levels. The collagen content levels in the tissue of various groups are shown in Figure 11.

### 2.12. Histopathology

Epicardium, endocardium, and myocardium, as well as papillary muscles and vasculature, were found to be in normal orientation in subjects treated with normal saline, which was judged to be a normal control. The toxic Control group (TCG) displayed infarction along with mural thrombi and occasionally acute aneurysms. The other groups were contrasted with the Toxic Control group (TCG), where the Sericin Standard group (SSG) demonstrated focal lesions without any evidence of myonecrosis while the Nanoparticle-treated Group (NTG) and Carvedilol Standard group (CSG) displayed normal, intact myocardial tissue with no infiltration. The cardiac tissue in the per se group displayed normal architecture, an orderly pattern of myofiber striations with central nuclei, and no vacuolization. Figure 12 displays images of histopathology.

## 3. Discussion

Nanotechnology, over the years, has not only improved the quality of pharmaceuticals but also the health and overall well-being of humans. Nature, the master architect of molecules created an infinite gathering of molecular entities. This plethora of molecular entities created by nature stands as a never-ending supply for drug development, new chemotypes, pharmacophores, and scaffolds for magnification into effective drugs for different ailments and other important bioactive agents. A promising candidate for therapeutic applications is a mixture of molecules with artificial and natural origins. Natural origin medicines have long been the foundation of folk medical systems throughout the world. They have also played a crucial role in history and culture [14]. Since tomotherapy has been used for as long as and frequently in conjunction with medicinal plants and presents a significant alternative to allopathic medicine in many parts of the world, these discoveries provide insight into the field of natural origin drug discovery, which primarily relies on drugs derived from plant species [15].

Sericin possesses good antioxidant properties, cardioprotective equity, flavonoids, and wound-healing properties. Various Unani preparations used in dealing with cardiovascular ailments have silk extract as their main ingredient [16].

Myocardial infarction caused by doxorubicin is a well-established model to comprehend the beneficial effects of several medicines and antioxidants on heart function. Cardiovascular diseases (CVDs), commonly known as heart illnesses, represent several conditions that affect the heart and blood vessels. According to estimates, 17.07 million people died from CVDs in 2015, accounting for 31 percent of all deaths that occurred globally. More people died from CVDs in 2015 than from any other disease [17].

When the myocardium suffers a metabolic injury, a large number of enzymes are released into the extracellular fluid (ECF) [18]. These signs of myocardial injury do not indicate how it happened, however [19]. The number of enzymes present in serum is directly connected to the total number of necrotic cells and is increased by the leaking of enzymes from cardiac tissue as a result of toxic-induced necrosis. Toxic substances may harm the myocardium by causing lipid peroxidation that is mediated by free radicals [18,20].

When compared to NCG rats, toxic group (TCG) animals showed a considerable increase in the activities of myocardial injury marker enzymes in serum along with a corresponding decline in myocardial tissue. The total concentration of the marker enzymes appeared to be lower in the cardiac tissue of TCG rats when compared to NCG, which may be a reflection of the effects of lipid-peroxide-induced cellular damage. In TCG rats, the increased levels of the serum indicator enzymes and their associated decrease in the cardiac tissue confirmed the beginning of myocardial necrosis [18].

Serum marker enzymes were discovered to be significantly lower in rats from the Nanoparticle Group (NTG), Carvedilol Standard Group (CSG), and Sericin Standard Group (SSG) than in the rats from the Toxic Group (TCG). According to reports, sericin’s antioxidant and free-radical-quenching properties may be the cause for these observations [16]. Sericin’s anti-free-radical capabilities may have reduced necrotic harm while also preventing enzyme seepage from the tissue. Sericin, a chief phenolic compound has been described to have strong scavenging activity and hold a protective effect contrary to myocardial ischemia and significantly reduces the release of serum AST, ALT, LDH, ALP, and CK [15]. When compared to the NCG, the PNG group revealed no significant increase in heart damage enzymes.

There is always a need for a separate isoenzyme study because the overall CK represents the sum of several individual isoenzyme portions. Therefore, an increase in overall CK does not identify the source of the CK. Creatine kinase isoenzyme MB (CK-MB) activity in plasma shows the persistence and magnitude of increases and is therefore helpful in estimating the extent of infarction [21]. This enzyme level dramatically rose when rats that had been exposed to harmful substances were compared to normal control rats. The presence of increased levels of marker enzymes such as creatine kinase-isoenzyme helped identify the heart abrasion caused by toxins (CK-MB). The extent of CK-MB activity is inversely correlated with infarct size [22]. Consequently, the treated group animals dramatically fell in number when compared to the toxic group.

A variety of cardiovascular disorders, including severe myocardial necrosis tissue, are frequently diagnosed using serum levels of cardiovascular troponins [23]. These effects of Dox on the heart are caused by the release of cTn from the myocardium, which increases serum cTnT just seven days after Dox treatment. The degree of myocardial damage is directly correlated with serum levels of cTn [24]. By stabilizing the cellular membrane, minimizing myofibril breakdown, and thus reducing troponin release in the blood, NTG, CSG, and SSG animals showed decreased troponin secretion in the current investigation.

The interaction between polyunsaturated fatty acids and the cell membrane, which encourages lipid peroxidation, may be the cause of the increased free radical formation [25] The levels of TBARS, HP, and CD in the heart and plasma of Dox-induced experimental rats TCG were significantly higher than the normal control [26]. Increased oxygen free radical production or decreased metabolism were the two possibilities for the rise in the amount of lipid peroxidative markers. Vitamin E is a lipid-solvable, chain-flouting antioxidant proficient in scavenging oxygen molecules, and centered free radicals [27]. According to human studies, there is a negative correlation between total plasma vitamin E levels and myocardial ischemia mortality [28]. Vitamin E effectively inhibits the lipid peroxidation autocatalytic process [29].

One of the body’s most prevalent non-enzymatic antioxidant biomolecules is GSH. In our research, rats treated with Dox (TCG) had lower GSH levels. Free radicals could be more difficult to eliminate with less GSH present, and this could result in a range of negative effects on the myocardium [30]. In rats receiving NTG and SSG treatment, there was an increase in GSH levels, which lowers the production of free radicals after cardiac damage.

Its ability to quench free radicals may be attributed to its flavonoid and polyphenol levels. This could be the cause of the increased amounts of vitamin C, vitamin E, and GSH found in the serum of rats given sericin and nanoparticle treatments. As a result, the nanoparticle’s antioxidative impact may be attributed to the observed increase in the levels of the vitamins E, C, and GSH in NTG rats.

The first line of cellular defense against oxidative stress is comprised of free radical scavenging enzymes such as SOD, CAT, GPx, GST, and GR. The efficient elimination of oxygen-mediated stress in intracellular organelles depends on the balance between these enzymes. The production of harmful reactive oxygen species (ROS) is observed from the outflow of electrons onto oxygen from a variety of systems throughout our entire body, and the endogenously acting antioxidant and enzyme defense are significantly involvedin preventing oxygen free-radical-facilitated tissue damage [31]. A thiol with a very low molecular mass called glutathione functions as the body’s main endogenous antioxidant and affects many different types of cells. It significantly contributes to the decline of disorders such as arteriosclerosis and reoxygenation injury [32].

The GPx, GST, GR, and CAT-SOD pair, which depend on GSH, are capable of removing harmful free radicals. Because glutathione is used more frequently to protect proteins with SH from lipid peroxides, glutathione levels may have decreased following the delivery of harmful agents. The activity of glutathione peroxidase and glutathione s-transferase on the administration of hazardous substances is also decreased when glutathione is less readily available [33]. When compared to rats in the hazardous group, the NTG, CSG, and SSG rats showed a notable boost in these enzymes. According to Deoriet al. (2016), sericin functions as an antioxidant at several stages of the oxidative process by preserving the balance between cellular and environmental oxidants. Because sericin has antioxidant properties, it may have been assumed in the current investigation that it prevented GSH oxidation by preserving the internal GSH antioxidant balance against damaging assisted cellular oxidation. This would have protected the GSH-associated enzymes [34].

Farvin typically demonstrated a significant decrease in the activity of SOD and catalase in rats from the toxic-control group (TCG) [35]. The involvement of hydrogen peroxide and superoxide free radicals in toxin-mediated cardiac cellular necrosis may be the cause of the decrease in SOD and catalase. To protect the tissue from damage caused by free radicals, the enzymes should have been utilized more. The creation of H_2_O_2_ and O_2−_, which naturally can lead to the formation of hydroxyl radicals (OH−), and can further disrupt the heart membrane, can result from a decrease in the activity of SOD and catalase enzymes [36].

Rats from the NTG, CSG, and SSG groups had significantly higher levels of SOD and CAT compared to the hazardous group. The antioxidant operates on the oxidation system at many different levels, and it may do so by lowering the localized O_2_ content, which produces phenoxy radicals and controls the polymer chain at its commencement by reacting with ROS to create freely reactive oxo radicals [37]. In rats treated with nanoparticles, SOD and catalase enzyme levels may have increased for this reason.

The current study discovered that an increase in the release of lysosomal hydrolases into the cytosol from the confined sacs was responsible for the breakdown of the cardiac membrane. This is about a previous theory that the cytosolic acid hydrolases released from lysosomes and those released from the sarcoplasmic reticulum cause the death and dysfunction of the sarcolemma, mitochondria, and other cellular organelles [38]. As a general indicator of the integrity of the lysosomal membrane, these catalysts, which operate like hydrolytic enzymes in blood circulation, can be measured [39].

Involved in energy metabolism and vulnerable to oxidative damage, mitochondria are significant subcellular organelles. The major organelles implicated in cardiac injury are mitochondria, which are abundant in cardiomyocytes [40]. This might affect both the cause and the result of oxidative cardiac damage. According to the literature, Dox promotes necrosis in the heart by lowering antioxidant potential, increasing the production of mitochondrial ROS, and altering the function of the mitochondrial respiratory chain by downregulating specific mitochondrial respiratory enzyme functions [41].

When compared to NCG rats in the current study, the effects of TCA cycle enzymes or mitochondrial respiratory chain enzymes such IDH, −KDH, SDH, and MDH were significantly diminished in the Dox-treated rats. The actions of these enzymes may be inhibited by ROS, which may have an impact on the mitochondrial substrate oxidation, lowering action transfer rates that are equal to molecular oxygen molecules, and catabolism of cellular energy [42]. Treatment with nanoparticles markedly increased the number of enzymes in NTG and SSG animals in comparison to the hazardous group. The per se group showed an increase in the number of mitochondrial enzymes in the myocardium when compared to an NCG.

The most prevalent protein in the extracellular cardiovascular matrix is collagen. To maintain the shape of the ventricles and distribute the contractile force from all myocardial cells to the ventricular canal, collagen fibers in the heart form a network of connecting myocytes. Maintaining the structure of cardiac tissue is crucial for maintaining the geometry and operation of the heart chambers. There are two types of collagen found in the heart: I and III. Because of its hard and tensile characteristics, collagen I in particular has a significant impact on ventricular stiffness [43].

Histopathology is entirely reliant on microscopic analysis and interpretation. Correct biopsy technique, suitable fixation and processing methods, suitable sectioning, and appropriate staining are fundamental prerequisites for arriving at a definitive diagnosis. For a thorough diagnosis, it is crucial to identify the structural and morphological characteristics of tissue components. The pathological changes such as apoptosiswere identified as indicated in the image [44,45].

## 4. Material and Methods

### 4.1. Reagents

Prepared lyophilized nanoparticles were used as test material. Doxorubicin and Sericin were purchased from Sigma Aldrich (St. Louis, MO, USA). Carvedilol was purchased from Yarrow chem products, Maharashtra, India. Other chemicals, reagents, and diagnostic kits of research/analytical quality were acquired from a nearby chemical source and employed in this work.

### 4.2. Preparation of Drug-Loaded Nanoparticles

According to Shariq et al., chitosan nanoparticles were created using the ionic gelation process. In a nutshell, chitosan was dissolved in a solution of 1% *v*/*v* acetic acid, then sericin 0.2% *w*/*v* was dissolved in chitosan solution directly and carvedilol 0.02% *w*/*v* was dissolved in ethanol 0.5 mL followed by adding to chitosan solution. Separately, water-dissolved tripolyphosphate was added dropwise to the chitosan solution while stirring, and the mixture was stirred for 10 min before being sonicated. The resulting nanoparticle formulation was lyophilized at −55 °C [46].

### 4.3. Experimental Animals

Adult Wistar rats (4-week-old) in good health weighing 150–180 g were obtained from the Central Drug Research Institute (CDRI) animal house facility located in Lucknow. All of the experimental rats were kept in propylene cages with a 12-h light/dark cycle and suitable housing circumstances, including a room temperature of 23–20 °C. The Institutional Animal Ethics Committee (IAEC) of the Faculty of Pharmacy, Integral University, Lucknow (U.P.), India, approved the study process with approval number (IU/IAEC/19/04), (Reg no.1213/PO/Re/S/08/CPCSEA, 5 June 2008).

Rats were randomly divided into 6 groups (*n* = 6). The doxorubicin-induced cardiotoxicity model was employed and the treatment protocol is given in Table 1.

### 4.4. Processing of Tissue and Blood Samples

To collect the plasma and serum, the blood was drawn into heparinized and non-heparinized tubes. The heart tissue was extracted, cleaned, weighed, and then homogenized. The supernatant was then collected and kept for further use.

### 4.5. Heart Weight Body Weight Ratio

The animals were euthanized, and their body weights were noted. The rat was pinned on its extended extremities and placed in a dissection tray. They were then washed with 70% ethanol to prevent hair and dandruff. A scalpel was used to cut into the chest, and the heart was removed and dried on blotting paper after being washed with normal saline. By dividing the heart weight by the total body weight, the heart-body weight ratio was obtained [47].

### 4.6. Cardiac Markers

AST, ALT, ALP, LDH, Creatinine Kinase, Creatinine kinase-myocardial band (CK-MB), and Troponin-T tests were performed using diagnostic kits.

### 4.7. Oxidative Stress Parameter

The malondialdehyde was assessed by the Thiobarbituric acid (TBA) test technique of Buege and Aust, (1978) [48]. Briefly, 0.4 mL of serum and 0.6 mL of TCA-TBA-HCl reagent were added. Blended and kept in the water bath for 10 min. After cooling, 1 mL of freshly prepared 1 N NaOH solution was added. The appearance of pink color was evaluated at 535 nm against blank. The technique described by Ohkawa et al., (1979) [49] was used to estimate the amount of lipid peroxide. In brief, 10,000 rpm for 10 min centrifugation was performed on 1 mL of 10% heart tissue homogenate. A 0.5 mL volume of 30% TCA and 0.5 mL of 0.8% TBA were added to this, which was then held at 80 °C for 30 min and centrifuged for 15 min at 3000 rpm. At 540 nm, absorbance was measured. The estimate method of Jiang et al. was used to measure the lipid hydroperoxide in the tissues and plasma (1992). In a nutshell, a mixture of 0.2 mL of sample and 1.8 mL of Fox reagent was incubated for 30 min at 37 °C, then centrifuged for 10 min at 10,000 rpm. At 540 nm, absorbance was detected [50]. The method described by Rao and Recknagel was used to investigate conjugated dienes (1968). Briefly, 5.0 mL of chloroform–methanol (2:1 *v*/*v*) reagent was combined with 0.1 mL of tissue homogenate or plasma. The mixture was then centrifuged for 5 min. The bottom layer was separated and evaporated to dryness by centrifugation. After adding 1.5 mL of cyclohexane, the absorbance was measured at 233 nm [51].

### 4.8. Antioxidant Parameters

Non-enzymatic antioxidants marker (serum/plasma): Ascorbic acid concentration was estimated using the method outlined by Aye Kyaw (1978) [52]. In brief, 2 mL of plasma and 2 mL of color reagent were combined, and the supernatant’s absorbance at 700 nm was measured in comparison to a blank solution made consisting of 2 mL of distilled water and 2 mL of a color reagent. For the determination of serum vitamin E shown by Baker and Frank (1968) [53], the colorimetric approach was employed. In a test tube, 1.5 mL of serum, 1.5 mL of ethanol, and 1.5 mL of xylene were combined thoroughly before being centrifuged at 5000 rpm for 10 min. After separating 1 mL of the xylene layer, the dipyridyl reagent was applied. At 460 nm, absorbance was measured. The method developed by Ellman G.L. in 1959 [54] was used to measure reduced glutathione (GSH). In short, 2.0 mL of 5% TCA and 2.0 mL of the sample were mixed before centrifugation, and subsequently 2.0 mL of the supernatant was removed. This was then mixed with 4 mL of 0.3 M disodium hydrogen phosphate and 1.0 mL of Elman’s reagent. At 412 nm, the emergence of the yellow color was assessed.

Non-enzymatic antioxidants marker (heart): The titration strategy described by Sadasivam and Manickam (1992) [55] was used to measure ascorbic acid. In a nutshell, 50 mg of fresh tissue was homogenized with 5 mL of oxalic acid, filtered, and then subjected to dichlorophenol indophenols titration. The amount of dye consumed was computed. The method described by Varley H (1976) [56] is used to determine -tocopherol (Vitamin E). In brief, 1.5 mL of heart tissue extract and 1.5 mL of xylene were combined, thoroughly blended, and centrifuged. After separating 1.0 mL of the xylene layer, the 2,2’-dipyridyl reagent was added and thoroughly mixed. A measurement of absorbance at 460 nm was made, and the amount of vitamin E was estimated. Boyne and Ellman (1972) calculated the decreased glutathione levels [57]. In short, 1.0 mL of 10% tissue homogenate was given 4.0 mL of metaphosphoric acid treatment. By centrifuging, the precipitate that had developed was removed. Disodium hydrogen phosphate and DTNB reagent were each added to 2.0 mL of supernatant. The amount of glutathione was determined using an absorbance measurement at 412 nm.

Superoxide dismutase (SOD) activity was examined using the method described by Marklund and Marklund in 1974 [58]. Enzymatic antioxidant marker (Heart): In a nutshell, 1.5 mL of Tris HCl and 1.0 mL of pyrogallol were combined with 1.0 mL of cardiac tissue homogenate. At 420 nm, absorbance was measured. Using the approach described by Aebi (1984) [59], the characteristics of catalase were assessed. Briefly, 4 mL of phosphate buffer and 20 μL of serum were combined, and the resulting mixture was then incubated at 25 °C. An additional 10 mM H2O2 was added to this solution, 0.65 mL total. At 240 nm, absorbance was measured. The method of Rotruck et al. was used to estimate the amount of GPx in tissue (1973) [60]. The following ingredients were added to 0.5 mL of tissue homogenate: 0.2 mL of Tris buffer, 0.2 mL of EDTA, and 0.1 mL of sodium azide. Additionally, 0.1 mL of hydrogen peroxide and 0.2 mL of glutathione were added to this solution, and the combination was then incubated at 37 °C for 10 min. Subsequently, 0.5 mL of a 10% TCA solution was added, and then centrifugation was performed for 10 min at 3000 rpm. A 1.0 mL volume of Elman’s (DTNB) reagent and 3.0 mL of disodium dihydrogen phosphate were added to 2.0 mL of the resultant supernatant. At 412 nm, absorbance was recorded and computed. GR concentration was evaluated using the Carlberg and Mannervik technique (1985) [61]. Briefly, 10 μL of tissue homogenate was mixed with 0.5 mL of phosphate buffer, 50 μL of GSSG, 50 μL of NADPH, and distilled water. At 340 nm, absorbance was measured and calculated. For the measurement of GST, the Habig et al. (1974) method was adopted [62]. The following ingredients were added, mixed, and incubated for 5 min, 1.0 mL of the sample, 2.7 mL of phosphate buffer, 0.1 mL of GSH, and 0.1 mL of CDNB. The absorbance was measured and computed.

### 4.9. Lysosomal Hydrolases

The method given by Conchie et al. (1967) [63] was used to analyze -glucosidase and -galactosidase. In brief, 0.5 mL of serum was combined with 2 mL of buffer, 0.5 mL of the substrate (o-nitrophenyl -galactoside for -galactosidase and o-nitrophenyl -glucoside for -glucosidase), and stored at 37 °C for 60 min of incubation. The liberation of nitrophenol (yellow color) was calculated by measuring the absorbance at 410 nm after the addition of 4.0 mL of glycine-NaOH solution. The estimate kit evaluated the -galactosidase assay. To estimate the activity of -galactosidase, 0.3 mL of serum extract was combined with 1 mL of buffer, 0.5 mL of the substrate (O-nitrophenyl-galactoside), and incubated at 37 °C for 1 h. After adding 3 mL of glycine–sodium carbonate buffer, the amount of p-nitrophenol that was liberated was calculated by measuring the absorbance at 410 nm. The procedure described by Barrett was used to test the cathepsin B action (1980). The activator reagent (2.0 mM disodium salt of EDTA and 2.0 mM cysteine hydrochloride in 0.14 M Na_2_HPO_4_/KH_2_PO_4_ buffer, pH 6.0) was applied to 100 μL of 10% tissue homogenate. BAPNA (50 μL) (40 mg/mL in DMSO) and the substrate were incubated at 37 °C for 1 h after 5 min of activation. In addition, 2.0 mL of 2 M Tris HCl buffer (pH 9.0) was added. At 410 nm, the development of yellow color was then measured [64]. The technique described by Sapolsky et al., was used to test the cathepsin D action (1973). Briefly, 1.0 mL of the substrate, 0.2 mL of enzyme homogenate, and 0.8 mL of 0.2 mM sodium formate buffer (pH 3.5) were combined and incubated at 37 °C for two hours. TCA was added to 2.0 mL of this 10% solution. centrifuged for 15 min at 3000 rpm. It was well mixed with 2.5 mL of sodium carbonate and a 4% solution of 0.1 NaOH. Then, the Folin’s reagent was added and blended. At 670 nm, color intensity was measured [65].

### 4.10. Heart Mitochondrial Enzymes

The modified techniques of Fontana-Ayoub M (2005) [66] were used to isolate the heart’s mitochondrial components. Briefly, 2 mL of isolation buffer B was combined with small fragments of heart tissue. After combining the tissue homogenate with 3 mL of isolation buffer B, the mixture was centrifuged at 800× *g* at 4 °C for around 10 min. After being separated, the supernatant was once more centrifuged at 10,000× *g* for approximately 10 min at 4 °C. They separated the supernatant. A 500 μL volume of isolation buffer A was used to resuspend the mitochondrial suspension. Once more, centrifugation was carried out for around 10 min at 10,000× *g* and 4 °C. 200 μL of suspension buffer was used to resuspend the mitochondrial suspension. A second 10,000× *g* centrifugation at 4 °C took place for around 10 min. In 200 μL of suspension buffer, the mitochondrial suspension was reconstituted. A 5 μL volume of the mitochondrial suspension was transferred to a 2 mL container and stored at −20 °C for subsequent examination to identify the mitochondrial enzymes. The method shown by Bell and Baron (1960) [67] was used to examine the catalyst (enzyme) isocitrate dehydrogenase (IDH) action. Briefly, 0.3 mL of manganous chloride, 0.2 mL of the substrate, 0.2 mL of mitochondrial suspension, and 0.4 mL of Tris-HCl buffer were added. This mixture was then incubated for 60 min followed by the addition of 1.0 mL of DNPH and then 0.5 mL of EDTA solution. After 20 min,10.0 mL of 0.4 NaOH was added. At 390 nm, the color’s intensity was measured. The Reed and Mukherjee (1969) technique was utilized to measure the enzyme’s ketoglutarate dehydrogenase activity [68]. Briefly, magnesium sulfate, thiamine pyrophosphate, potassium ferricyanide, and -ketoglutarate were combined with 0.15 mL of phosphate buffer. A 0.2 mL volume of mitochondrial suspension was added to this solution, which was then incubated at 30 °C for 30 min. Additionally, 1 mL of 10% TCA was added before centrifuging. After separating the supernatant, 0.1 mL of potassium ferricyanide, 0.5 mL of ferric ammonium sulphate–dupanol reagent, and 1 milliliter of 4% dupanol were added. The mixture was then heated at 25 °C for 30 min. At 540 nm, absorbance was measured. *Succinate dehydrogenase (SDH)* enzymatic assay was evaluated by a technique used by Slater and Bonner (1952) [69]. A 0.2 mL volume of mitochondrial suspension was added after 0.1 mL of EDTA, 0.1 mL of bovine albumin, 1 mL of phosphate buffer, 0.2 mL of potassium ferricyanide, 0.3 mL of sodium succinate, and 0.1 mL of potassium cyanide were combined. At 420 nm, color intensity was measured and computed. Mehler et al. (1948) [70] examined the function of malate dehydrogenase. Briefly, 0.1 mL of mitochondrial suspension was added after 0.3 mL of buffer, 0.1 mL of oxaloacetate, and 0.1 mL of NADH had been combined. At 340 nm, color intensity was measured and computed.

### 4.11. Tissue Collagen Content

A 0.2 mL volume of mitochondrial suspension was added after 0.1ml of EDTA, 0.1 mL of bovine albumin, 1 mL of phosphate buffer, 0.2 mL of potassium ferricyanide, 0.3 mL of sodium succinate, and 0.1 mL of potassium cyanide were combined. At 420 nm, color intensity was measured and computed. Mehler et al. (1948) [64] examined the function of malate dehydrogenase. Briefly, 0.1 mL of mitochondrial suspension was added after 0.3 mL of buffer, 0.1 mL of oxaloacetate, and 0.1 mL of NADH had been combined. At 340 nm, color intensity was measured and computed. After cooling in tap water for 5 min, 1.0 mL of freshly produced PDAB (a 20% solution in methyl cellosolve that had been warmed to 60 °C to aid in solubilization) was added, thoroughly mixed, and then placed in a water bath for 20 min at 60 °C. At 557 nm, the acquired color was measured spectrophotometrically against a blank made of distilled water. To create a calibration curve, standard solutions were handled similarly. By dividing the hydroxyproline concentration by 8, the collagen content of the tissue samples was calculated and expressed as mg/gm tissue [71].

### 4.12. Histopathology

Heart tissue was processed for histopathological slides, and pictures were taken in accordance with protocol. Following euthanasia, the hearts and lipids were carefully removed. They were promptly fixed in 10% buffered formalin for 48 h, dried by progressive immersion in various water–ethanol concentrations, washed with xylene, and then again embedded in paraffin before being sectioned into 5–6 μm sections with the aid of a microtome. Haematoxylin and eosin dyes, among others, were used to stain the sections [44,45].

### 4.13. Statistical Analysis

Statistical analysis was carried out on GraphPad Prism 8.0.1. by performing one-way ANOVA followed by Dunnett’s test.

## 5. Conclusions

According to the results of the current study, oral nanoparticle administration at a dose of 200 mg/kg has a strong cardioprotective effect against experimental myocardial necrosis and hypertrophy caused by doxorubicin, lowering oxidative stress and inflammatory reactions that result in noticeably improved myocardial activity and attenuated heart damage after myocardial ischemia. Individual doses of carvedilol and sericin were shown to be less efficacious than the effect of nanoparticles. The results of the previous study revealed a novel method for treating heart diseases that involves a combination regimen with one synthetic and one natural component. Because of its effectiveness and safety, nanoparticle ingestion may one day be regarded as a cardioprotective medicine for patients. Additionally, this study offers fresh perspectives on how different combination formulations might be developed to function as targeted combination therapy for cardiovascular disorders.

## Figures and Tables

**Figure 1 pharmaceuticals-16-00561-f001:**
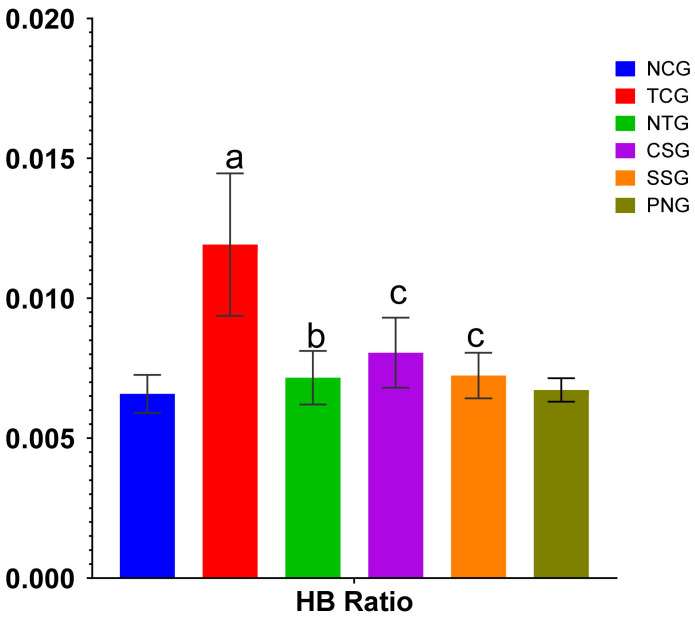
Heart: Body weight ratio for different treatment groups. The presented values for all variables were mean ± SD (*n* = 6). TCG and PNG were compared to Normal Control, and Treatment groups (NTG, CSG, and SSG) were compared to TCG, the degree of significance was found to be ^a^
*p* < 0.001, ^b^
*p* < 0.01, and ^c^
*p* < 0.05.

**Figure 2 pharmaceuticals-16-00561-f002:**
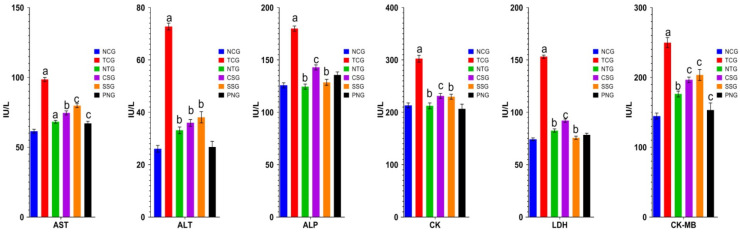
Serum Biochemical estimations of various treatment groups. The presented values for all variables were mean ± SD (*n* = 6). TCG and PNG were compared to Normal Control, and Treatment groups (NTG, CSG, and SSG) were compared to TCG, the degree of significance was found to be ^a^
*p* < 0.001, ^b^
*p* < 0.01, and ^c^
*p* < 0.05.

**Figure 3 pharmaceuticals-16-00561-f003:**
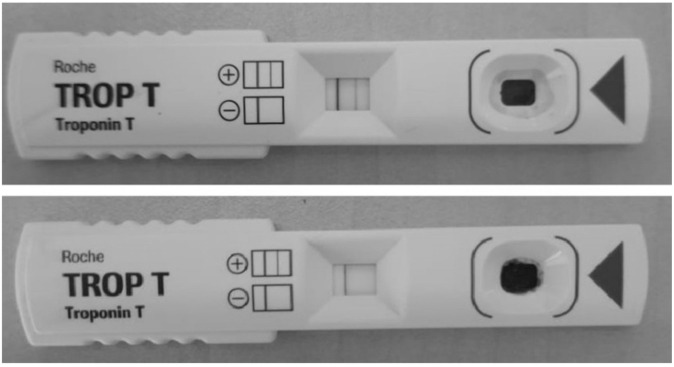
Troponin T test showing positive as well as negative test results.

**Figure 4 pharmaceuticals-16-00561-f004:**
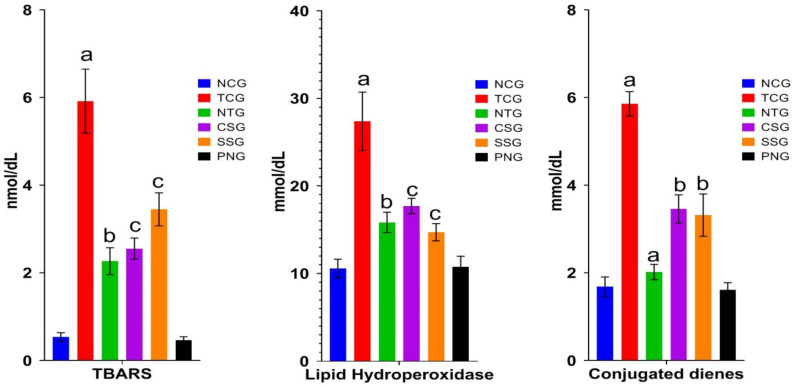
Lipid peroxidative markers of various treatment groups. The degree of significance was found to be ^a^
*p* < 0.001, ^b^
*p* < 0.01, and ^c^
*p* < 0.05.

**Figure 5 pharmaceuticals-16-00561-f005:**
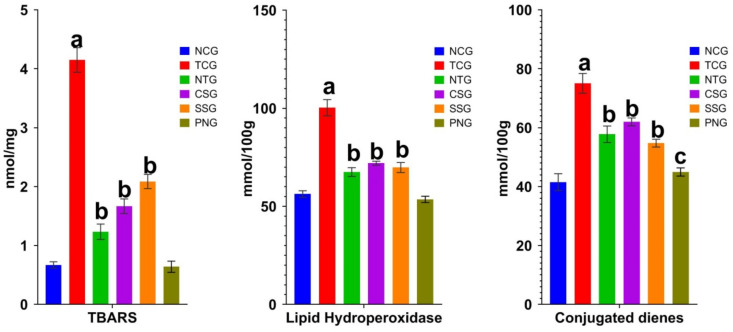
Lipid peroxidation markers level viz. TBARS, Lipid Hydroperoxide and Conjugated dienes (heart) of various treated groups. The degree of significance was found to be ^a^
*p* < 0.001, ^b^
*p* < 0.01, and ^c^
*p* < 0.05.

**Figure 6 pharmaceuticals-16-00561-f006:**
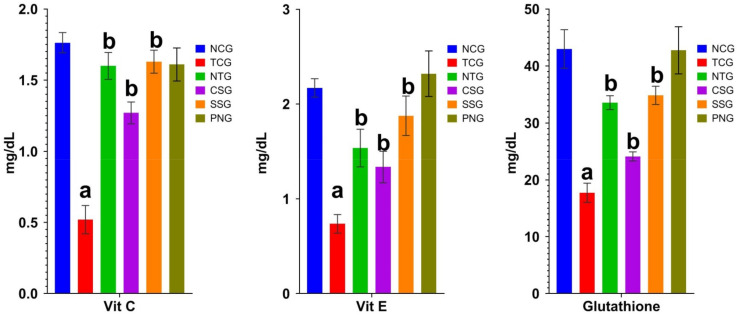
Non-enzymatic antioxidants markers level viz. Vitamin C, Vitamin E and Glutathione (Plasma) in the various treatment groups. All data were reported as mean ± Standard deviation (*n* = 6). Treatment groups (NTG, CSG, and SSG) were compared to TCG, while TCG and PNG were compared to Normal Control, and the levels of significance were observed as ^a^
*p* < 0.0001 and ^b^
*p* < 0.001.

**Figure 7 pharmaceuticals-16-00561-f007:**
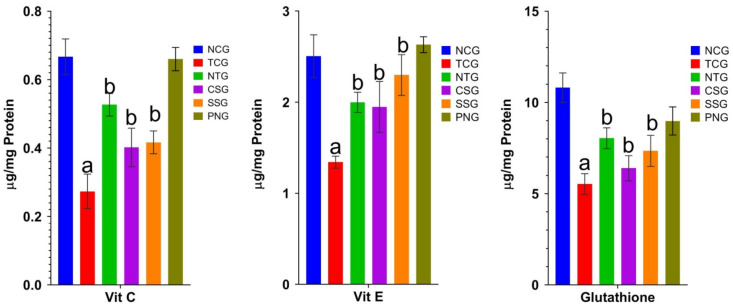
Non-enzymatic antioxidant marker levels (Heart) in the different treatmentgroups. All data were reported as mean ± Standard deviation (*n* = 6). Treatment groups (NTG, CSG, and SSG) were compared to TCG, while TCG and PNG were compared to Normal Control, and the levels of significance were observed as ^a^
*p* < 0.0001 and ^b^
*p* < 0.001.

**Figure 8 pharmaceuticals-16-00561-f008:**
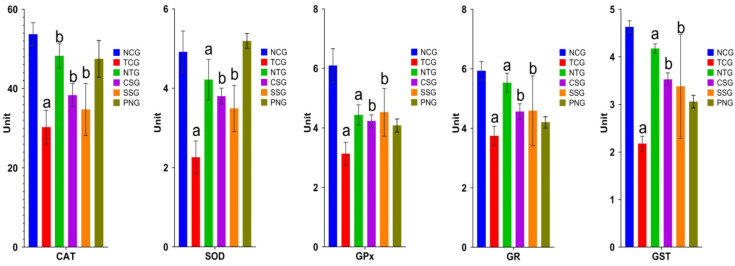
Enzymatic antioxidant marker levels (Heart) in the different treatment groups. All data were reported as mean ± Standard deviation (*n* = 6). Treatment groups (NTG, CSG, and SSG) were compared to TCG, while TCG and PNG were compared to Normal Control, and the levels of significance were observed as ^a^
*p* < 0.0001 and ^b^
*p* < 0.001. The graph was plotted using the Log10 scale.Units: SOD: One unit of enzyme activity was defined as the enzyme reaction that prevented 50% of nitroblue tetrazolium (NBT) reduction in one minute. CAT: μmole of hydrogen peroxide decomposed/min/mL.GPx: μmole of GSH utilized/min/mg protein; GST: μg of CDNB conjugate formed/min/mg protein.GR: μg of reduced glutathione formed/min/mg protein.

**Figure 9 pharmaceuticals-16-00561-f009:**
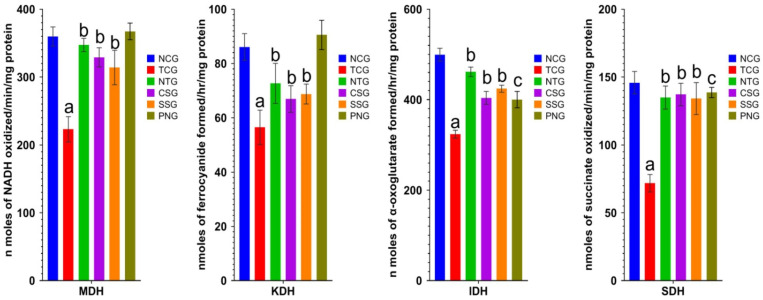
Mitochondrial enzyme levels (heart)in the different treated groups. All values were reported as mean ± Standard deviation (*n* = 6). Treatment groups (NTG, CSG, and SSG) were compared to TCG, while TCG and PNG were compared to Normal Control, and the levels of significance were observed as ^a^
*p* < 0.0001, ^b^
*p* < 0.001 and ^c^
*p* < 0.01. Unit of IDH: n moles of α-oxoglutarate formed/h/mg protein; α-Ketoglutarate Dehydrogenase: nmoles of ferrocyanide formed/h/mg protein; Succinate Dehydrogenase: nmoles of succinate oxidized/min/mg protein; Malate dehydrogenase: n moles of NADH oxidized/min/mg protein.

**Figure 10 pharmaceuticals-16-00561-f010:**
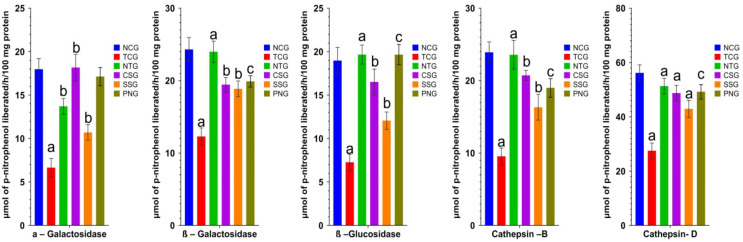
Lysosomal hydrolases level (heart) in the different treated groups. All data were reported as mean ± Standard deviation (*n* = 6). Treatment groups (NTG, CSG, and SSG) were compared to TCG, while TCG and PNG were compared to Normal Control, and the levels of significance were observed as ^a^
*p* < 0.0001, ^b^
*p* < 0.001 and ^c^
*p* < 0.01.

**Figure 11 pharmaceuticals-16-00561-f011:**
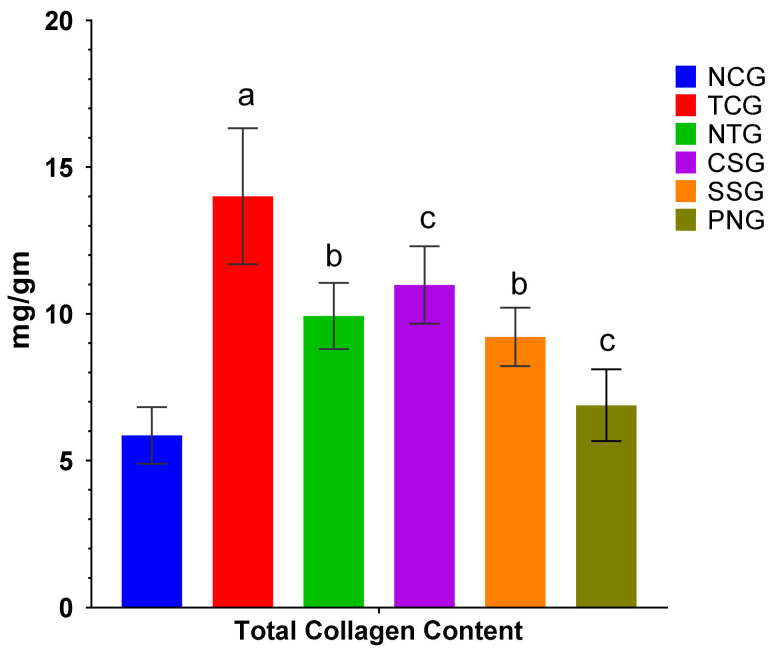
Total collagen content (heart) in the different treated groups. All data were reported as mean ± Standard deviation (*n* = 6). Treatment groups (NTG, CSG, and SSG) were compared to TCG, while TCG and PNG were compared to Normal Control, and the levels of significance were observed as ^a^
*p* < 0.0001, ^b^
*p* < 0.001 and ^c^
*p* < 0.01.

**Figure 12 pharmaceuticals-16-00561-f012:**
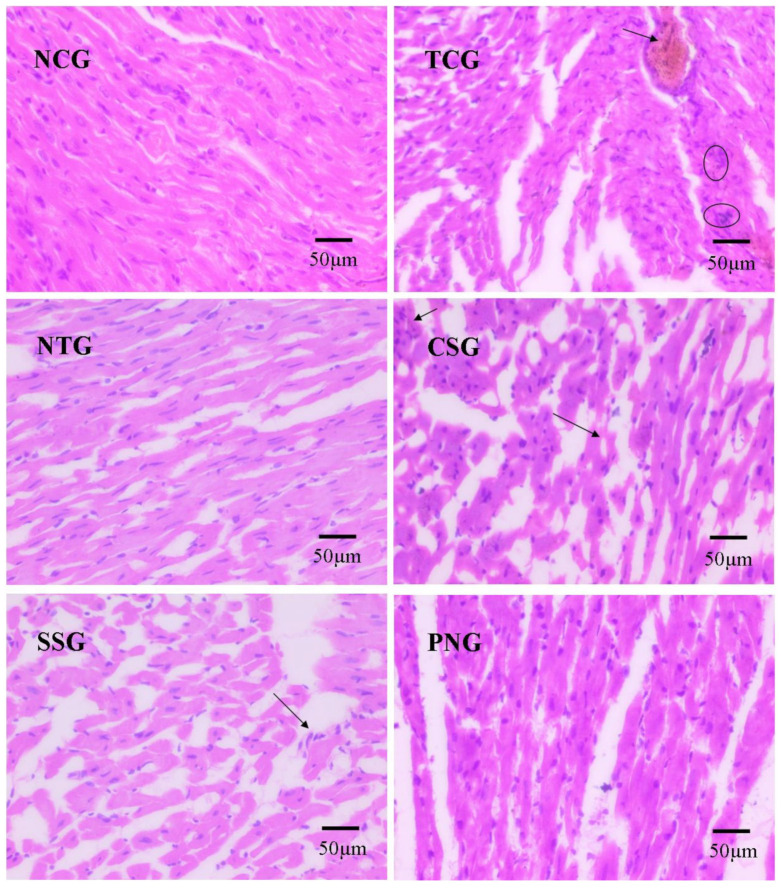
Histopathological images of the heart in different treatment groups. Arrows and circle in TCG shows necrosis and infiltration respectively. Arrows in CSG and SSG shows vacuolization.

**Table 1 pharmaceuticals-16-00561-t001:** Treatment protocol provides details about treatment dose, route, and duration.

Groups	Treatment	Dose, Route, and Duration
Normal Control Group (NCG)	Normal Saline (NS)	2 mL/kg/p.o/day for 7 days
Toxic Control Group (TCG)	Dox + NS	Dox 20 mg/kg i.p. once on 1st day + DW 2 mL/kg/p.o/day for next 6 days.
Nanoparticle Treated Group (NTG)	Doxorubicin + Nanoparticle	200 mg/kg p.o. from day 1 to day 7 + Dox 20 mg/kg i.p. once on 1st day after 2 h of Nanoparticle administration
Carvedilol Standard Group (CSG)	Doxorubicin + Carvedilol	10 mg/kg p.o. from day 1 to day 7 + Dox 20 mg/kg i.p. once on 1st day after 2 h of Carvedilol administration
Sericin Standard Group (SSG)	Doxorubicin + Sericin	100 mg/kg p.o. from day 1 to day 7 + Dox 20 mg/kg i.p. once on 1st day after 2 h of Sericin administration
Per Se Group (PNG)	Nanoparticle	200 mg/kg p.o. for 7 days

## Data Availability

Data are contained within the article.

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
