# Peer review of "Fabrication of Nanoformulation Containing Carvedilol and Silk Protein Sericin against Doxorubicin Induced Cardiac Damage in Rats"

_pharmaceuticals, 2023, doi:10.3390/ph16040561_

Round 1

Reviewer 1 Report

Article Title: “Fabrication of Nanoformulation containing Carvedilol and silk protein Sericin against doxorubicin-induced cardiac damage in rodents”

Reviewer: Dr. Timur Saliev

Date: 07-10-22

Comments:

The authors investigated the efficiency of cardio-protective combination of nano-formulation of sericin and carvedilol.

The abstract is concise and well written.

The Introduction chapter is written very nicely.

The methods and study design are appropriate and clearly described.

The results are well provided and discussed in Discussion section.

The English language is a high quality.

References are relevant and appropriate.

Some comments/suggestions:

Please provide information for  NTG, CSG, SSG (etc.) abbreviation in Abstract, beginning of the manuscript and in the legends for Figures as well.

Please add p-values as asterisks and bars to the all graphs (instead of letters)

To summarize, I would recommend accepting the manuscript after minor revision.

Author Response

A rebuttal file has been uploaded

Reviewer 2 Report

In this article,  M.Shariq et al., described cardioprotective effects of nano formulation containing carvedilol and sericin against doxorubicin induced cardiac toxicity. The authors have performed series of biochemical estimations (mostly related to oxidative stress) in both plasma as well as in heart and showed that the nano formulation attenuated the Dox induced oxidative stress.

The work is very superficial and lacks the rationale and clear experimental approach. The results are neither represented nor descried in a proper, scientific way. The following are some of my concerns that authors may consider to improve the quality of their work.

1.     The biggest setback for this study is that authors have fail to assess any cardiac functional parameter to demonstrate the actual benefit of their intervention (nano formulation). Mere alterations in the antioxidant enzymes, doesn’t necessarily prove the therapeutic efficacy of the intervention. The clinical success of anti-oxidants is limited in most of the cases including the cardiovascular diseases. Hence, it is important for the authors to show some sort of favourable cardiac functional alteration with the intervention to value their claims.

2.     From the manuscript, I failed to understand the reason for administering the two drugs in a nano formulation. What’s the reason for enclosing them as a nano formulation? Poor solubility, absorption? Please elaborate this in this introduction section so that the readers understand the rationale for the study.

3.     The representation of multiple parameters in the same graph with a common X-axis and Y-axis should be avoided. It’s hard to understand the numerical values for some parameters and make sense of their group comparisons.

4.     Although use of abbreviated group names is encouraged, they need to be elaborated and the treatment conditions need to be described in detail for each group in figure legends.

5.     The phenotypic data for the study needs to be discussed. The body weight changes, heart weigh/body weight, heart weight/tibial length, lung weights need to be shown in the manuscript.

6.     Authors need to provide a brief method for each of the biochemical estimation they have performed instead of just citing the reference from which they have extracted the method.

7.     In the results section, each figure should be clearly described and the fold changes or percentage changes (numbers and the p values) should be used. For instance in the results section for the mitochondrial enzymes, authors have just mentioned that they have obtained the changes in the mitochondrial enzymes as shown in the figure 8, instead of clearly explaining how they were altered with Dox and then co-treatment with nano formulation.

8.     The entire manuscript needs to be revised for the English correction.

Author Response

A rebuttal file has been uploaded.

Reviewer 3 Report

The manuscript submitted for publication relates to the development of a nanoformulation containing carvedilol and the silk protein sericin against doxorubicin-induced heart damage in rodents.

The authors of this article evaluated the cardioprotective potential of a novel nanoformulation of sericin and carvedilol. By determining the levels of various serum biochemical markers of myocardial damage, the authors confirm that the developed nanoparticle formulation is effective against doxorubicin-induced cardiotoxicity.

Throughout, in the introduction, the authors analyzed relevant literature data related only to Dox-induced toxicity. It is necessary to add a few more sentences regarding the cardioprotective effects of the new nanoformulation used in this study.

The results are well-written with a concise explanation.

All tables and figures were well organized, clear, and with all the necessary information.

Materials and methods are good and fully explained.

The discussion is very well written.

A significant number of references support the presentation of the study, but as I stated in the comments given directly in the text of the article, certain parts of this manuscript must be supplemented with adequate and recent references.

Also, authors must correct English grammar and spelling.

I believe that the article can be accepted for publication after "minor revision" in accordance with my suggestions (please see the attached PDF document of the article).

Author Response

A rebuttal file has been uploaded.

Reviewer 4 Report

Dear author, 

I hope you are well and safe while I write this. The current study evaluates the potential cardioprotective effects of novel sericin and carvedilol nanoformulations. The author measured serum cardiac biomarkers and lipid peroxide levels while evaluating the new formula in doxorubicin (Dox) induced cardiotoxicity (Plasma and heart tissue). The author's findings suggested that the new formulation had a successful outcome against cardiotoxicity caused by doxorubicin (Dox).

I recommend the author revise the following:

1-     Revise the article for missing spaces.

2-     Add an informative sentence about the animal used in the current study.

3-     In the result section in the Troponin T part, the author should indicate the statistical significance of the result.

4-     In the result section, Mitochondrial Enzymes (Heart Tissue)Lysosomal Hydrolases (Heart Tissue) and, Total Collagen Content, the author did not describe the results at all. Descriptive comments s with statistical inference are required.

5-     In line 254, author stated “The per se group demonstrated”, what did you mean?

6-     The used abbreviation or acronym with writing the full word even for the first time of mentioning.

7-     The author did not mention the characteristics of the used nano formulation even the proportion of formulation is not mentioned 

8-     In line 375, The author did not mention the dose used to induce doxorubicin cardiotoxicity

Yours

Author Response

A rebuttal file has been uploaded.

Round 2

Reviewer 2 Report

The revised version of this article by M.Shariq et al., addressed some of my concerns. However, I still feel that this article misses important functional, mechanistic aspects.

1.     The manuscript still lacks functional aspects. Authors need to show some sort of cardiac functional measure (echocardiography or ECG changes) that are altered by Dox and further changes with nano formulation.

2.     I am not sure how changes in expression patterns of myosin heavy chains corresponds to the apoptotic signalling? Authors need to run western blots for caspases, PARP, Bax, Bcl2 proteins to show influence of Dox and co-treatment with nano formulations on apoptotic signalling.

3.     The histology images shown in the manuscript doesn’t represent the characteristic features of Dox induced cardiac damage such as myocyte loss, matrix disorganisation, inflammatory cell accumulation etc.

4.     I still don’t understand author’s logic behind having multiple parameters with common x and y axes. They say it’s for sub parameter comparison. But, they don’t need to be cross compared. Moreover, it’s difficult to understand the numerical values for some parameters.  For instance, what’s the value of TBARS for NCG and PNG groups in Fig.5?

5.     The units of each parameter need to be displayed on Y-axis, instead of just writing Values.

6.     Dunnet’s test is a posthoc test for multiple comparisons, it’s not a t test.

Author Response

The various concerns of the reviewer have been keenly observed and rectified in revised manuscript.

Reviewer 4 Report

I hope this letter finds you healthy and secure. The current study assesses the cardioprotective potential of new nanoformulation of sericin and carvedilol. The author evaluated the new formula in doxorubicin (Dox) induced cardiotoxicity and assessed serum myocardial biomarkers and lipid peroxide levels (Plasma and heart tissue). The author's results implied that the new formulation had a good outcome against doxorubicin (Dox) induced cardiotoxicity.

I recommend the acceptance of the manuscript in its current form.

YOurs

Author Response

The motivating words of reviewer have encouraged us to improve the quality of our research in our future publications.

Round 3

Reviewer 2 Report

No comments

Author Response

The various improvements have been incorporated into the manuscript as indicated by the reviewer. The language and grammar of the manuscript have been improved using Grammarly software.